# Eat Your Broccoli: Oxidative Stress, NRF2, and Sulforaphane in Chronic Kidney Disease

**DOI:** 10.3390/nu13010266

**Published:** 2021-01-18

**Authors:** Scott E. Liebman, Thu H. Le

**Affiliations:** Division of Nephrology, Department of Medicine, University of Rochester School of Medicine and Dentistry, 601 Elmwood Avenue, Box 675, Rochester, NY 14642, USA; Scott_Liebman@urmc.rochester.edu

**Keywords:** chronic kidney disease, oxidative stress, Nrf2, sulforaphane, GSTM1

## Abstract

The mainstay of therapy for chronic kidney disease is control of blood pressure and proteinuria through the use of angiotensin-converting enzyme inhibitors (ACE-Is) or angiotensin receptor blockers (ARBs) that were introduced more than 20 years ago. Yet, many chronic kidney disease (CKD) patients still progress to end-stage kidney disease—the ultimate in failed prevention. While increased oxidative stress is a major molecular underpinning of CKD progression, no treatment modality specifically targeting oxidative stress has been established clinically. Here, we review the influence of oxidative stress in CKD, and discuss regarding the role of the Nrf2 pathway in kidney disease from studies using genetic and pharmacologic approaches in animal models and clinical trials. We will then focus on the promising therapeutic potential of sulforaphane, an isothiocyanate derived from cruciferous vegetables that has garnered significant attention over the past decade for its potent Nrf2-activating effect, and implications for precision medicine.

## 1. Introduction

Chronic kidney disease (CKD) affects about 14% of the United States population [1]. Individuals with CKD disease have a high risk of cardiovascular morbidity and mortality, and this risk increases with CKD severity [2]. In addition to the adverse clinical course, CKD care is expensive and consumes a disproportionate amount of health care dollars [3]. Treatment of CKD has historically focused on addressing traditional risk factors such as hypertension, proteinuria, and, in diabetics, blood glucose control [4]. While these treatments can delay disease progression, many patients with CKD ultimately progress to end-stage kidney disease and require renal replacement therapy.

Oxidative stress—a state of imbalance between the generation and degradation of free radical oxidant compounds, including reactive oxygen species (ROS) and reactive nitrogen species (RNS)—is increasingly recognized as an important factor in the initiation and progression of chronic kidney disease [5,6,7]. Commonly generated endogenous ROS and RNS include the superoxide (O_2_^.−^), hydroxyl (OH^.^), nitric oxide (NO^.^), and nitrogen dioxide (NO_2_^.^) radicals. At physiologic concentrations, ROS and RNS fulfill homeostatic roles such as cell signaling and the synthesis of some cell structures, and are used in phagocytic cell defense against pathogens [8]. The pathophysiologic effects occur when there is an imbalance between oxidation and reduction—an altered redox state in which excess free radicals react with other molecules, including lipids, proteins, and nuclear DNA. Lipid peroxidation damages cell membranes and lipoproteins, leads to formation of toxic reactive aldehydes, and promotes further lipid peroxidation, ultimately affecting a large number of lipid molecules [8]. When ROS react with proteins, they may induce conformational changes that render the proteins partially or completely nonfunctional [8]. ROS reaction with DNA may lead to mutagenesis, disruption of the cell cycle, and induction of apoptosis [8,9]. This review will focus on oxidative stress in chronic kidney disease, and sulforaphane, a bioactivator of the NRF2 pathway, as a potential therapy to mitigate this stress.

## 2. Sources of Oxidative Stress in the Kidney

In the kidney (as in other tissues), ROS and RNS may be produced via a variety of mechanisms. The kidneys (in particular, the proximal tubules) require a significant amount of energy in the form of adenosine tri-phosphate (ATP) in order to achieve solute reabsorption, protein synthesis and degradation, and regulation of glomerular filtration required to maintain homeostasis [10]. Mitochondria are the cellular organelles that provide the major source of ATP via the electron transport chain. In this reaction, electrons are transferred between a series of electron donors and acceptors, ultimately to oxygen, which is then reduced to water [10,11]. This process, called oxidative phosphorylation, generates energy and drives the generation of ATP via an ATP synthase [10]. During electron transfer, some of the reactions may be incomplete, leading to the premature leakage of free electrons which can interact with oxygen and generate excess ROS [12]. Similar to lipids, proteins, and nuclear DNA, mitochondrial DNA is also susceptible to oxidative damage. This may result in impaired transcription of mRNA and post-translational modification of mitochondrial proteins, including oxidative phosphorylation enzymes and proteins involved in antioxidant defense, ultimately resulting in both ROS generation and compromised ATP production [10]. Mitochondrial dysfunction has been shown to play a major role in progression of renal diseases, including acute kidney injury and diabetic nephropathy [13].

NADPH (nicotinamide adenine dinucleotide) oxidases (NOX) are another potential source of oxidative stress. NOX catalyze the oxidation of NADPH or NADH and produce the ROS species O_2_^−^ and H_2_O_2_, which react with iron ions Fe^3+^ and Fe^2+^, respectively, in the Haber–Weiss reaction, resulting in the net generation of •OH and OH^−^ radicals [14]. At low basal activity, NOX generate ROS that have physiologic functions in cell proliferation, metabolism, and death. However, under conditions such as activated renin angiotensin system (RAS) or a high-salt diet, NOX activity in the kidney is upregulated [15] and can lead to overproduction of H_2_O_2_ and other free radicals with deleterious consequences.

Endothelial nitric oxide synthase (eNOS) produces nitric oxide (NO) via the oxidation of L-arginine to L-citrulline in a process requiring tetrahydrobiopterin (BH_4_) as a cofactor [16]. NO has a critical role in vascular biology by mediating a cell signaling cascade that leads to vasodilation, thereby lowering blood pressure. The bioavailability of NO can be increased by polyphenols found in plant-based foods that can augment eNOS expression, as well as by nitrates found in high concentrations in many leafy green vegetables and beets that serve as an eNOS-independent source of NO [17]. Under pathophysiologic conditions, eNOS can act as another source of free radicals by generating O_2_^.−^ instead of NO in a process known as eNOS uncoupling [18]. The importance of eNOS in kidney health is illustrated in a study showing that deletion of eNOS in the mouse resulted in progressive renal abnormalities, including tubular apoptosis and necrosis, and glomerular and tubular scarring [19].

Other enzymatic sources of free radicals include myeloperoxidase (MPO), which normally catalyzes the H_2_O_2_-mediated oxidation of halide ions, but under pathophysiologic conditions oxidizes other substrates and mediates tissue damage [20]. In a renal ablation model of CKD, MPO knockout mice displayed attenuated glomerular injury and decreased expression of markers of fibrosis and inflammation [21], suggesting that increased activity of MPO may contribute to kidney disease progression. Xanthine oxidase, in the process of generating uric acid, generates oxidants such as O_2_^.−^ and H_2_O_2_ [22]. In addition, there is evidence supporting the role of uric acid in hypertension [23] and CKD progression, in part through its direct role activating NADPH oxidase [24,25].

Free radicals and oxidative stress may also be generated via exogenous sources such as radiation, xenobiotic compounds, cigarette smoke, and environmental or industrial toxins such as heavy metals or solvents [7,26].

To combat oxidative stress and prevent pathophysiologic alterations in the redox state, organisms have developed a number of antioxidant defenses. Antioxidants inhibit the formation of free radicals or mitigate their damaging effects by: (1) acting as an alternative substrate for oxidation, thereby preventing an oxidation reaction which would lead to the formation of free radicals, (2) directly scavenging free radicals, or (3) indirectly preventing the development of oxidant compounds by either upregulating antioxidant defenses or inhibiting free radical production [27]. These antioxidants may be endogenous or exogenous from food sources [9], and can be enzymatic or nonenzymatic. The enzymatic antioxidants may be primary and constitutively acting such as superoxide dismutase (SOD), catalase (CAT), and peroxidase. Superoxide dismutase catalyzes the reaction of O_2_^−^ to H_2_O_2_ and O_2_ [28]_._ The H_2_O_2_ may be subsequently converted to oxygen and water via CAT or via a peroxidase such as glutathione peroxidase (GPx), used to oxidize another substance such as gluthathione [GSH] whose thiol groups allow them to donate electrons to detoxify the free radicals [28]. Alternatively, oxidative stress may alter the activity of transcription factors mediated by antioxidant responsive elements (AREs) in promoter regions of genes that combat oxidative stress [29]. The nuclear factor erythroid 2 related factor 2 (NRF2) is one such critical transcription factor that is discussed in more detail below.

## 3. Evidence That Oxidative Stress Is Operant in CKD

There are several lines of evidence establishing a link between oxidative stress and CKD. Some studies have shown that individuals with CKD have elevated markers of oxidative injury [30,31,32,33] that are, depending on the marker evaluated, proportional to the degree of kidney dysfunction [30,31,33]. Mechanistically, mitochondrial dysfunction, as discussed above, is an increasingly recognized contributor, particularly in diabetic nephropathy, in which mitochondrial ROS production exceeds the local antioxidant capacity [13,34]. Evidence suggests that mitochondrial dysfunction may also play a role in other kidney diseases, including IgA nephropathy, membranous nephropathy, and polycystic kidney disease [35]. All of the mechanisms discussed above have been shown to be present in CKD, namely upregulated NOX activity, eNOS uncoupling, and pro-oxidative activities of myeloperoxidase and xanthine oxidase [7,12]. Moreover, a number of studies showed when levels of antioxidant agents such as SOD, CAT, GPx/glutathione, and, as discussed in more detail below, NRF-2 are reduced, the harmful effects of oxidation and generation of ROS cannot be appropriately mitigated [12].

## 4. NRF2 in Kidney Disease

Nuclear factor erythroid 2 related factor 2 (NRF2) plays a central role in protecting cells from oxidative injury. As a transcription factor, NRF2 induces gene expression of enzymes that combat the effects of oxidative stress [36]. NRF2 target genes share a common DNA sequence known as the antioxidant response element (ARE) in their promoter regions that is required for NRF2 binding and gene induction [37]. NRF2 is constitutively expressed in the cytoplasm of all cell types; however, it is normally kept at low levels via Kelch-like ECH-associated protein 1 (KEAP1)-mediated ubiquitination and degradation. This also serves to keep NRF2 target antioxidant genes at the low, basal levels needed to maintain their “housekeeping” functions [38]. When modified during periods of oxidative stress, interaction between reactive oxygen species and cysteine residues of KEAP1 allows NRF2 to escape KEAP1-mediated ubiquitination and degradation and translocate to the nucleus where it can induce ARE-containing targets for the purpose of restoring oxidative homeostasis in the cell [38]. The myriad target genes include antioxidant proteins, phase I oxidation, reduction, and hydrolysis genes, phase II detoxifying enzymes such as glutathione s-transferases (GSTs), NADPH-generating enzymes, drug transporters, and stress proteins involved in heme and metal metabolism, such as heme oxygenase 1 (HO-1) [26].

The role of NRF2 has been studied in both animal models and human CKD. In animal models, under normal, healthy conditions, NRF2 knockout mice exhibit no abnormalities throughout their lifespan. However, in disease state models, such as cardiac disease, diabetes, and obesity, loss of NRF2 augments disease severity [36]. In the kidney, studies using both genetic and pharmacologic approaches have revealed the protective effect of NRF2 in animal models of CKD.

Jiang et al. studied the role of NRF2 in a streptozotocin (STZ)-induced diabetic nephropathy model in *Nrf2* wild-type (WT) and knockout (KO) mice [39]. After 16 weeks, despite a similarly achieved level of hyperglycemia, *Nrf2* KO mice had a higher degree of oxidative damage, more albuminuria, and more severe glomerulosclerosis, compared to WT mice [39]. In other models of kidney disease, including autoimmune nephritis, toxic injury, ischemia reperfusion injury (IRI), ureteral obstruction, and podocyte injury, *Nrf2* KO organisms also displayed an increase in disease severity, suggesting that NRF2 plays a nephroprotective role through a common pathway [39,40,41,42,43,44,45,46,47,48,49,50]. Similar nephroprotection is seen with NRF2 activators and KEAP1 suppressors (which allows NRF2 to translocate to the nucleus and exert its effect) [36]. In mouse models of ischemia and unilateral ureteral obstruction, KEAP1 hypomorphic mice displayed attenuated kidney disease compared to KEAP1-intact mice [47]. Zheng et al. studied the role of the NRF2 activators sulforaphane (more below) and cinnamic aldehyde in a STZ-induced diabetic nephropathy model in *Nrf2* wild-type and knockout mice. They found that the NRF2 activators attenuated markers of kidney damage and minimized glomerular pathology in wild-type but not in *Nrf2* knockout mice [50]. However, *Nrf2* deletion has been shown to be beneficial in a model of autoimmune nephritis, by increasing sensitivity to tumor necrosis factor-alpha (TNF-α)-mediated apoptosis [51]. Similarly, in the Akita mouse model of Type 1 diabetes, genetic deletion or pharmacological inhibition of Nrf2 attenuated hypertension and kidney disease [52]. It is possible that effect of NRF2 is disease context-dependent. Nevertheless, taken together, the data suggest that NRF2 has a nephroprotective role in kidney disease.

In humans, the most well-studied pharmacologic agent activating the NRF2 system is the drug bardoxolone methyl, which covalently binds to cysteine residues of KEAP1, allowing NRF2 to escape ubiquitination and degradation and translocate to the nucleus to induce the myriad antioxidant genes [53]. The potential beneficial effect of bardoxolone methyl on kidney function was first observed in a phase I cancer trial, where it was found to result in a statistically significant increase in estimated glomerular filtration rate (eGFR) of 26% [54]. Based on this finding, bardoxolone methyl was investigated as a potential treatment for CKD. The earliest study evaluated 20 patients with stage 3b–4 chronic kidney disease (eGFR range 15–45 mL/min/ 1.73 m^2^) due to diabetes. In this non-placebo-controlled trial, eGFR statistically increased after 4 and 8 weeks compared with baseline (2.8 mL/min/1.73 m^2^ and 7.2 mL/min/1.73 m^2^ at each time point, respectively) [55].

This study was followed by the BEAM study—a double-blinded, placebo-controlled trial designed to assess the efficacy and safety of bardoxolone methyl in patients with Stage 3b to 4 diabetic kidney disease. Of the 227 patients enrolled, 57 received placebo, and the remainder were evenly divided into three different doses of bardoxolone methyl (25 mg, 75 mg, and 150 mg). After one year, the change in eGFR was significantly higher in the bardoxolone methyl groups as compared with placebo (5.8 ± 1.8 mL/min/1.73 m^2^ for 25 mg vs. placebo, 10.5 ± 1.7 mL/min/1.73 m^2^ for 75 mg vs. placebo, and 9.3 ± 1.9 for 150 mg ml/min/1.73 m^2^ vs. placebo) [56].

The BEACON trial was a phase 3 double-blinded, placebo-controlled trial in individuals with stage 4 diabetic kidney disease (eGFR range 15–29 mL/min/1.73 m^2^) designed to test the hypothesis that bardoxolone methyl would reduce the risk of end-stage kidney disease or death from cardiovascular causes in these patients [57]. The trial was stopped early due to an increased incidence of hospitalization or death from heart failure in the bardoxolone methyl group [57]. After a median of nine months of follow-up, however, the bardoxolone methyl group did show an increase in eGFR of 5.5 mL/min/1.73 m^2^ vs. −0.9 mL/min/1.73 m^2^ for the placebo group [57]. Post hoc analysis identified those at risk for fluid overload [58], and the TSUBAKI study evaluated whether bardoxolone methyl would increase eGFR in patients with diabetic kidney disease in whom these risk factors were absent [59]. In this study, 85 patients with stage 3 or 4 diabetic kidney disease (eGFR range 15–60 mL/min/1.73 m^2^), and with no identified risk factors for heart failure, were randomized to receive bardoxolone methyl or placebo. After 16 weeks of follow-up, the bardoxolone methyl group showed an increase in inulin-measured GFR of 5.95 mL/min/1.73 m^2^ vs. −0.69 mL/min/1.73 m^2^ for the placebo group.

The role of bardoxolone methyl in nondiabetic kidney disease is currently being evaluated in the CARDINAL study of Alport’s syndrome. Although not published at the time of this review, a press release from Reata pharmaceuticals highlighted the year one results: at 48 weeks of treatment, patients treated with bardoxolone had a statistically significant improvement in mean eGFR of 9.50 mL/min/1.73 m^2^ (*p* < 0.0001) compared to placebo [60].

Taken together, these data suggest that pharmacologic intervention in the KEAP1/NRF2 pathway can increase GFR. Whether this leads to a reduction of progression to end-stage kidney disease is still not certain, and the negative cardiovascular effects are still a concern. Further, despite the increase in GFR, bardoxolone methyl seems to lead to an increase in urinary protein excretion [57,59], raising the question and concern whether this could be related to the undesirable glomerular hyperfiltration or increased intraglomerular pressure that contributes to kidney disease progression long term [61]. An alternative explanation is that bardoxolone may have a favorable effect on glomerular surface area, and the increase in albuminuria may be tubular rather than glomerular in origin [62]. Animal studies suggest that bardoxolone has a narrow therapeutic window, since its metabolites could also be toxic [63]. Thus, the jury is still out regarding the long-term effect of bardoxolone on kidney function and disease progression.

## 5. Potential Role for Sulforaphane (SFN) in Kidney Disease

Nutrition has an important role in health and disease; certain nutrients are not only essential but they are also important for optimal health. Traditionally, food is viewed as a source of energy which, through its nutrients, can maintain homeostasis. However, it is now recognized that certain types of foods, known as functional foods, can provide more than just energy and essential nutrients [64]. Functional foods are those which contain bioactive compounds such as phytochemicals that have antioxidant and anti-inflammatory properties. As the hallmark of CKD is oxidative stress and inflammation, these compounds may play an important role in ameliorating or preventing CKD progression [64,65]. There are a large number of phytochemicals which can provide oxidative defense via upregulation of NRF2 [66]. Among these are glucosinolates (β-thioglucoside N-hydroxysulfates), including glucoraphanin found in all members of the plant family Brassicaceae, which encompasses a large number of commonly consumed species in the *Brassica* genus [67], including the species *Brassica oleracea*—better known as cruciferous vegetables such as broccoli, cauliflower, cabbage, kale, Brussels sprouts, kohlrabi, and collard greens [68].

Dietary administration of broccoli seeds induced a significant increase in activities of the phase II detoxification enzymes NAD(P)H dehydrogenase, quinone 1 (NQO1), and GSTs (that are downstream targets of Nrf2) in stomach, small intestine, and liver tissues of wild-type mice, but not in *Nrf2* KO mice [69]. The increased GST activities were associated with increased protein levels of GSTA1, GSTA2, GSTA3, and GSTM1 [69]. These studies illustrate that broccoli seeds induce antioxidant and detoxification proteins in a NRF2-dependent manner.

Many studies have shown the benefits of cruciferous vegetables intake including a decrease in: all-cause mortality [70,71,72,73], cardiovascular mortality [74,75], the incidence of type 2 diabetes [76,77], the incidence of renal cell carcinoma [78], and mortality in individuals with breast [79] and lung [80] cancer. To date, there are no data examining the association between cruciferous vegetables and outcomes in those with CKD. However, studies looking at plant protein as a whole have shown benefits of increased plant protein intake including a decrease in incident CKD [81,82], and a decrease in rate of decline in GFR in older women [83]. A recent analysis of data from the Chronic Renal Insufficiency Cohort found that, in those with prevalent CKD, adherence to a “healthy dietary pattern”, of which high vegetable intake is a prominent component, was associated with a lower risk of progression and mortality [84]. However, as several cruciferous vegetables such as Brussels sprouts and broccoli are also rich in potassium, the excretion of which becomes impaired with advanced CKD, patients are often advised to limit their consumption to prevent potentially life-threatening hyperkalemia. The beneficial effect of cruciferous vegetables on human health is thought to be in large part mediated by sulforaphane (SFN) [85,86]. Supplementation with SFN may therefore provide kidney-protective effects in those prone to hyperkalemia.

In the digestive tract, when a cruciferous plant tissue is initially injured by the process of mastication, glucoraphanin, a chemically inert glucosinolate, is exposed to the released enzyme myrosinase found in the plant tissue, or in the gut microbiome, and becomes hydrolyzed [86]. The result of this hydrolysis is the liberation of glucose and the formation of SFN and other products. SFN is a relatively small, lipophilic molecule (MW = 177.29 g/mol) which offers an advantage in bioavailability as compared to other larger phytochemicals such as hydrophilic polyphenols [85]. Animal studies using oral or intravenous SFN show that its bioavailability is up to ~80% [87]. Human studies of whole foods or supplements show that the bioavailability of SFN varies, and depends on many factors, including the type and part of cruciferous vegetable consumed, and their stage of maturation. For example, a study reported the glucoraphanin content in the seeds of a number of *Brassica* cultivars [88], which is summarized in Figure 1A. Another study examined the ability of extracts of different *Brassica* species to induce the quinone reductase NQO1 in a Hepa 1c1c7 murine hepatoma cell line [89] (Figure 1B). These studies illustrate that broccoli has the highest glucoraphanin content and antioxidant capacity. Different parts of broccoli also contain different levels of glucoraphanin and therefore exhibit different degrees of antioxidant effects. Broccoli seeds and sprouts contain the highest amount of glucoraphanin, and their extracts exhibit more antioxidant activity than broccoli heads [86]. Not surprisingly, consumption of broccoli strains with more glucoraphanin leads to higher plasma levels of SFN compared to consumption of strains with lower glucoraphanin levels [90].

The method by which broccoli is processed and prepared also affects the SFN content. Cooking broccoli changes the amount of SFN [91,92], as heat may destroy the enzyme myrosinase required for the conversion of the chemically inert glucoraphanin to the bioactive SFN [93]. Among the different cooking methods, stir frying and steaming retain more SFN than boiling [92], whereas microwaving can either increase or decrease SFN levels, depending on timing (shorter times can increase levels) and power setting [93,94,95,96]. Blanching and freezing broccoli to extend shelf life can also decrease the amount of SFN [97,98]. SFN levels can be increased if cooked broccoli is consumed with an exogenous source of myrosinase (such as powdered mustard seed) [99].

The most well-studied role of SFN is its action on the KEAP1-NRF2 pathway. By chemically modifying the cysteine sensors of KEAP1, thereby releasing NRF2 from KEAP1-mediated ubiquitination, SFN is one of the most potent NRF2 activators [100,101]. SFN is 13.5-fold and 105-fold more effective than curcumin and resveratrol, respectively, in inducing NQO1 [101]. Several preclinical studies suggest SFN may be a potential novel therapy for kidney disease.

In a mouse model of diabetic nephropathy induced by STZ, SFN attenuated hyperglycemia, polyuria, and polydipsia in wild-type mice, but not in *Nrf2* KO mice [50]. Compared to STZ treatment alone, addition of SFN decreased nephromegaly, glomerular collagen accumulation, glomerulosclerosis, glomerular basement membrane thickness, and urine albumin excretion, but only in wild-type mice [50]. Furthermore, wild-type, but not *Nrf2* KO, mice displayed increased protein levels of both NQO1 and γ-glutamylcysteine synthetase (γ-GCS), when treated with SFN [50]. A separate study using STZ-induced diabetic nephropathy in rats also demonstrated that SFN offered biochemical and histologic protection, as well as protection of DNA from oxidative damage [102]. In addition, SFN-treated animals also demonstrated reduced levels of mRNA expression of transforming growth factor beta (TGF-β) 1, collagen IV, and fibronectin, as well as decreased protein levels when assessed via immunohistochemistry [102]. SFN also increased transcriptional activation and protein levels of the antioxidant enzymes NQO1 and HO-1 [102]. In a rat model of STZ-induced diabetic nephropathy with superimposed contrast media injury, treatment with SFN partially abrogated renal injury, lowered renal markers of oxidative stress [malondialdehyde (MDA) and 8-hydroxy-2′-deoxyguanosine (8-OHdG)], and improved renal function [103]. SFN also offered protection in a model of type 2 DM in which mice were fed a high-fat diet followed by STZ injection [104]. In this study, SFN treatment mitigated the proteinuria and fibrosis seen in the diabetic mice, and also attenuated the kidney levels of profibrotic mediators (TGF-β and connective tissue growth factor), inflammatory mediators [plasminogen activator inhibitor -1 (PAI-1) and vascular cell adhesion molecule-1 (VCAM-1)], and indicators of oxidation (3-nitrotyrosine and 4-hydroxy-2-noneal) [104]. Taken together, these data indicate that SFN is protective in animal models of diabetic nephropathy, and its effect is mediated by activation of NRF2. Another study provided more evidence of SFN’s protective role by replicating the renal protective effect and NRF2 activation by SFN in mice with STZ-induced diabetes, and demonstrating that benefits were not sustained following three subsequent months without further treatment [105]. Expression of products of NRF2 activation such as NQO1 and HO-1 were significantly higher during SFN treatment, but this difference dissipated once treatment was stopped [105]. These data suggest continued SFN administration is needed to maintain the activation of the NRF2 pathway to confer protection against the oxidative damage of diabetes.

The renal protective effect of SFN has been demonstrated in many other models of kidney injury. In a mouse model of calcium oxalate nephrocalcinosis-induced kidney injury, SFN treatment decreased renal calcium oxalate deposition, inflammation, and cell death [106]. Gene expression analysis showed SFN-induced protection was associated with increased levels of NRF2 as well as decreased levels of toll-like receptor 4, a protein which upregulates the proinflammatory nuclear factor kappa β (NF-kB) [107]. In a rat model of unilateral ureteral obstruction (UUO), treatment with SFN attenuated pathophysiologic and histologic changes [108,109]. In addition, SFN partially improved renal blood flow and cortical and medullary O_2_ tension, and reduced the level of renal ROS [108]. Furthermore, SFN treatment increased nuclear NRF2 levels, and simultaneously decreased the level of cytoplasmic NRF2, suggesting SFN increased nuclear NRF2 activity [108]. SFN treatment also decreased renal infiltration of monocytes and macrophages, expression of proteins associated with cell death (from apoptosis, autophagy, or pyropotosis), and staining for TGF-β [108]. In a mouse model of obesity-related glomerulopathy, SFN-treated mice displayed a lower degree of albuminuria and improved glomerular architecture as compared to those not treated with SFN [110]. As in previous models, these changes were seen only in wild-type mice, but not in *Nrf2* KO mice, suggesting that the protective effect of SFN is mediated by NRF2 [110]. In a maleic acid acute kidney injury and acquired Fanconi syndrome model, pretreatment of rats with SFN prior to maleic acid injection abrogated increases in proteinuria and markers of proximal tubular injury and oxidative stress [111]. Moreover, mitochondrial oxidative phosphorylation capacity, which was reduced in maleic acid AKI, was restored in SFN-treated animals [112]. In a rat model of chronic renal allograft dysfunction, treatment with SFN delayed the rise in serum creatinine and proteinuria and attenuated the histologic findings of chronic allograft nephropathy [113]. Compared to untreated rats, those treated with SFN displayed lower renal cortical tissue levels of the oxidative stress markers malondialdehyde and 8-OHdG and higher levels of the antioxidant enzymes SOD, CAT, and GPx [113]. This protective effect was again likely mediated via activation of NRF2 [113]. A separate study using a transplant model in rats demonstrated that SFN improved post-transplant BUN and creatinine, and attenuated post-transplant tubular injury on light microscopy and mitochondrial shape and ultrastructure on electron microscopy [114]. Kidneys from SFN-treated animals also had higher levels of the antioxidant enzyme SOD-2 than kidneys from untreated ones [114]. In spontaneously hypertensive stroke-prone rats (SHRSP) [115], treatment with SFN significantly reduced blood pressure and significantly increased DNA methylation (an epigenetic mechanism to control gene expression [116]) to levels seen in control normotensive Sprague Dawley rats [115]. In addition, SFN reduced the number of sclerotic glomeruli in the SHRSP rats, and significantly decreased wall thickness and increased luminal area in small renal arteries and arterioles [115].

In rodent models of ischemia reperfusion injury [117,118], arsenic-induced nephropathy [119], hemolysis-mediated acute kidney injury [120], and gentamicin [121] and ochratoxin [122] induced nephrotoxicity, treatment with SFN ameliorated oxidative stress and cell death; attenuated the inflammatory cytokines interleukin-1β and interleukin-6 and several proapoptotic factors; and improved renal function [117,118,119,120,121,122].

The beneficial effects of SFN have also been illustrated in vitro. In human kidney 2 (HK2) cell culture, hippuric acid-mediated fibrosis was accompanied by an alteration of the redox state due to disruption of the NRF-2-driven antioxidant system [123]. Treatment of the cells with SFN significantly suppressed hippuric acid-mediated ROS and H_2_O_2_ production, and upregulated the antioxidant proteins NRF2, HO-1, and NQO1 [123]. SFN also decreased expression levels of the fibrosis-associated proteins collagen-I, alpha-smooth muscle actin, and vimentin [123]. In an HK2 cell culture model of hyperglycemia, treatment with SFN prevented high glucose-induced epithelial-to-mesenchymal transition, likely via an NRF2-dependent mechanism [124]. Pretreatment of HK2 cells with SFN protected against hypoxia–reoxygenation-induced cytotoxicity in a dose-dependent manner, and significantly augmented mRNA expression of phase 2 enzymes [118]. In other kidney cell lines (human embryonic kidney 293, human proximal tubule epithelial, and LLC-PK1 cells) exposed to cisplatin [125,126], SFN protected against a cisplatin-induced increase in ROS and decrease in antioxidant mediators [126] and against cisplatin-induced cell death [125]. SFN was also shown to be beneficial in attenuating uranium-induced cell damage in a rat kidney line (NRK-52) [127].

While the major protective mechanism by SFN is thought to be via NRF2 and induction of phase II enzymes, there are also other pathways potentially influenced by SFN, including the cytochrome P450 enzymes, apoptotic pathways, cell cycle progression, angiogenesis, and anti-inflammatory activity [128]. SFN also covalently modifies the N-terminal proline residue of macrophage migration inhibitory factor (MIF) [129] and inhibits its catalytic tautomerase activity [130]. SFN is also metabolized into several active metabolites: SNF–N-acetylcysteine (SFN-NAC), SFN-glutathione (SFN-GHS), and SFN-cysteine (SFN-Cys) [131]. Several of these metabolites have anticancer activities by modulating ERK1/2 phosphorylation [132] and microtubule function [132,133] although their biological actions have not been well studied.

Importantly, the strength of the beneficial effect of SFN may be determined by the presence or absence of the enzyme glutathione S-transferase µ-1 (GSTM1), a downstream target of NRF2. GSTM1 has been shown to have activity against several reactive aldehydes [134,135,136] and epoxides [137]. In humans, a common deletion variant of the *GSTM1* gene, the *GSTM1* null allele (*GSTM1(0)*), results in decreased or absent GSTM1 enzymatic activity and is associated with higher levels of oxidative stress. The highly prevalent *GSTM1(0)* is associated with more rapid CKD progression in the African American Study of Kidney Disease (AASK) trial participants [138]. This association has been replicated in the Atherosclerosis Risk in Communities (ARIC) study [139]. Recently, using a hypertensive model of kidney injury, Gigliotti et al. showed that *Gstm1* KO mice displayed augmented kidney injury and inflammation, compared to wild-type mice. Dietary supplementation of sulforaphane-rich broccoli powder ameliorated kidney disease only in *Gstm1* KO mice [130]. Similarly, in the ARIC study, high intake of cruciferous vegetables was associated with lower risks of kidney failure, with stronger effects in those homozygous for the null allele (*GSTM1(0/0)*), compared to those carrying one or two copies of the active allele [130]. It is likely that the bioavailability of SFN is influenced by GSTM1, as free SFN and SFN metabolites are increased in *GSTM1* null subjects compared to those with the active allele [140]. Importantly, in the context of personalized and precision medicine, the study by Gigliotti et al. highlights diet–gene interactions in kidney disease, and illustrates that response to disease-modifying effects of a nutrient such as SFN may be influenced by genetics.

As noted above, enthusiasm for bardoxolone has been tempered by potential cardiovascular adverse effects. As SFN activates the same NRF2 pathway, it follows that SFN could also pose similar adverse effects, particularly if taken in an isolated preparation rather than via cruciferous vegetables. Cell culture data suggest differential action of bardoxolone and SFN. A recent study evaluated the effects of bardoxolone and SFN on human dermal microvascular endothelial cell function [141]. Both agents augmented NRF2 expression, but bardoxolone was 2–5 times more potent in NRF2 activation than SFN at equal concentration [141]. Both agents decreased ROS production; however, bardoxolone was found to have more cytotoxicity. Endothelial cells incubated with 3, 5, or 10 µM bardoxolone showed a decrease in cell viability due to an increase in both apoptosis and necrosis, whereas endothelial cells treated with similar SFN concentrations did not display cytotoxicity [141]. Bardoxolone, but not SFN, also had detrimental effects on mitochondria as evidenced by a significant increase in proton leakage, and decrease in spare respiratory capacity and mitochondrial membrane potential [141]. It should be noted that, compared to control, SFN at “low dose” or 0.5 µM concentration, ex vivo, is able to induce expression of the genes in human peripheral blood mononuclear cells (PBMCs) that encode the enzymes Aldo-Keto Reductase Family 1 Member C1 (AKR1C1) and NQO1 that play a role in reducing radical species [142]. Higher doses of 2 µM and 5 µM of SFN induce higher expression of AKR1C1 and NQO1, as well as HO-1 [142]. In human trials, SFN has been reported to be well tolerated at peak plasma concentrations of 1–2 µM [143].

## 6. Conclusions

In summary, there is mounting supporting evidence that SFN may have therapeutic potential in kidney disease by stimulating the NRF2 pathway (Figure 2). SFN is currently in clinical trials for cancer of the breast, lung, and prostate, as well as autism and schizophrenia (Clinicaltrials.gov). However, currently no clinical study has been performed to assess the effect of SFN in CKD. As SFN and its metabolites are cleared by the kidney [144], safety and efficacy should first be established in patients with kidney disease, particularly in those with more advanced stages of CKD. Once established, large, randomized, placebo-controlled trials are needed to determine the effect of SFN on long-term outcomes such as disease progression and mortality in patients with CKD, and whether the effect is modified by genetics in a precision medicine approach.

## Figures and Tables

**Figure 1 nutrients-13-00266-f001:**
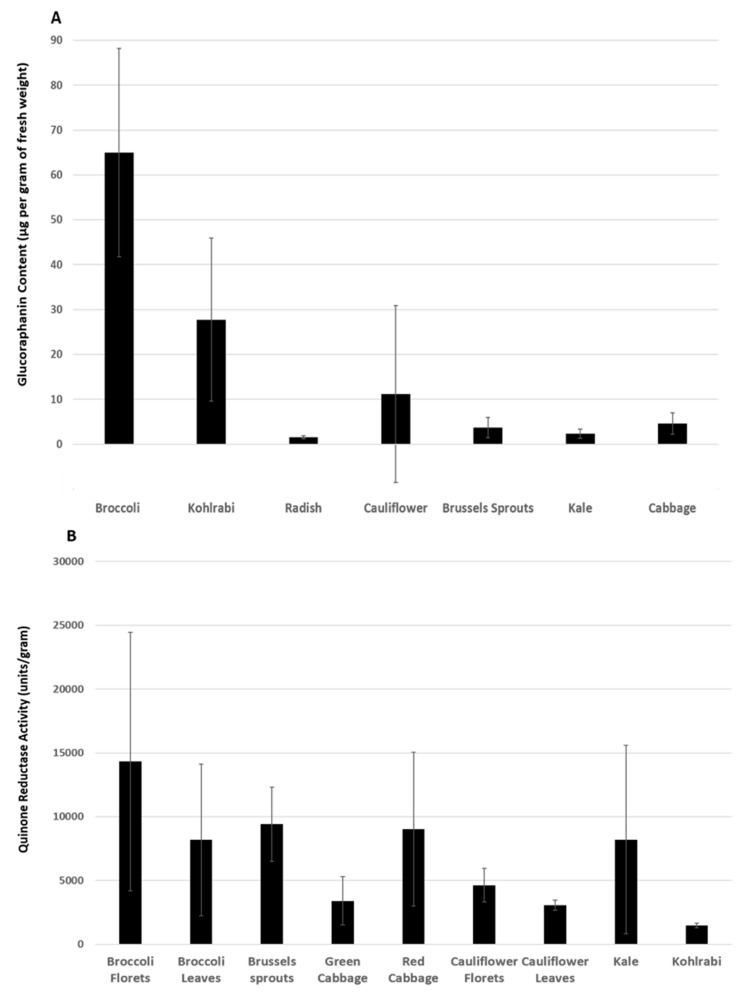
(**A**) Glucoraphanin content in the seeds of various Brassica vegetables. Data summarized from West et al. [88]. (**B**) Antioxidant ability of extracts of various *Brassica* vegetables, measured as quinone reductase activity induced in Hepa 1c1c7 murine hepatoma cells. Data summarized from Prochaska et al. [89].

**Figure 2 nutrients-13-00266-f002:**
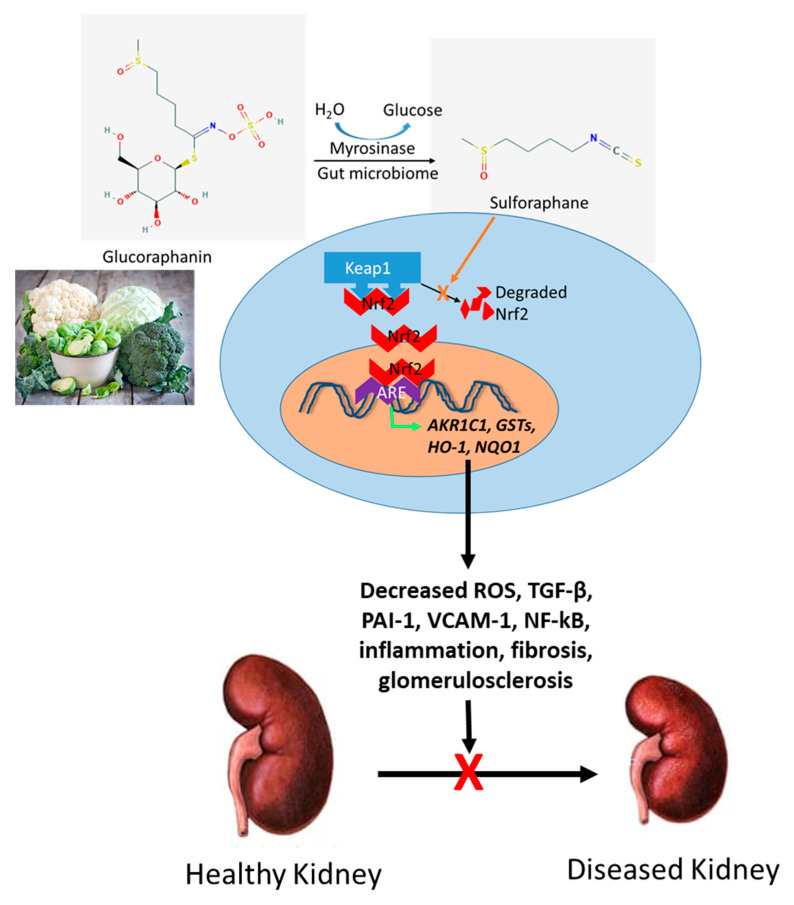
Sulforaphane targets Nrf2 pathway to prevent kidney disease progression. Glucoraphanin in cruciferous vegetables is hydrolyzed to sulforaphane by the enzyme myrosinase or by the gut microbiome. Sulforaphane inhibits the KEAP-1-mediated degradation of Nrf2, allowing Nrf2 to migrate from the cell cytoplasm into the nucleus where it binds to the antioxidant responsive element (ARE) in the promoter regions of Nrf2 target genes, thereby increasing their transcription, and hence their activities. The result is a decrease in ROS and factors involved in inflammation and fibrosis. Image of cruciferous vegetables taken from https://www.eatright.org/food/vitamins-and-supplements/nutrient-rich-foods/the-beginners-guide-to-cruciferous-vegetables. Image of healthy and diseased kidney taken from https://www.nephrologyspecialistsoftulsa.com/chronic-kidney-disease.php.

## Data Availability

Not applicable.

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
