# Peer review of "Eat Your Broccoli: Oxidative Stress, NRF2, and Sulforaphane in Chronic Kidney Disease"

_nutrients, 2021, doi:10.3390/nu13010266_

Round 1

Reviewer 1 Report

Definitely this is an interesting topic in the field of the kidney disease. The authors may focus more on the content of SFN and its mechanism - activation of NRF2 in the paper (as stated in the title and conclusion).

1) "Oxidative stress" and its pathway NRF2 in CKD are not new and are not the focus of this paper. The authors may consider to use simple diagrams to summarize what we already know for the introduction of this review.

2) Yes, the evidence of NFR2 in CKD is important. And the mechanism of SFN in the activation of NFR2 is known.

3) How much of SFN we found in Broccoli? how about in broccoli seed sprout and other sources/vegetables?

4) What are the challenges in the use of these functional foods for preventing CKD?

Author Response

 We thank the reviewer for positive and constructive critiques

  • We have added Figure 2 to summarize this review.
  • We have added Figures 1A and 1B to illustrate the levels of glucoraphanine and bioactivity of the various cruciferous vegetables.
  • We discussed the challenges of hyperkalemia that may complicate the consumption of these functional foods in CKD.
  • We have also made extensive revision of the review to expand and focus more on SFN
  • All changes are in red font.

Reviewer 2 Report

The present review focused on the therapeutic potential of sulforaphane in relations to chronic kidney disease (CKD). This isothiocyanate compound, derived from cruciferous vegetables, has the potent Nrf-2 activating effect. The role of Nrf-2/Keap1 in CKD development, as well as the potential mechanism, by which sulforaphane could counteract this process have been well described.
In my opinion, diagram showing the beneficial effect of sulforaphane on Nrf-2/Keap1 system in kidney tissue, including its anti-inflammatory, anti-oxidant and anti-fibrotic properties, would be helpful in understanding the role of sulforaphane in combating of CKD progression.

Author Response

We thank the reviewer for the positive comment and suggestion. We have added a diagram (Figure 2) to illustrate the key points of the review. We have also made extensive revision of the review to expand and focus more on SFN.  All changes are in red font.

Reviewer 3 Report

The review article "Eat your Broccoli: Oxidative Stress, NRF2, and Sulforaphane in Chronic Kidney Disease" by Scott E. Liebman and Thu H. Le  has tried to review literature on a well established and important antioxidant mechanism in CKD. Role of Nrf2 is well established in multiple oxidative stress related diseases such as asthma, sepsis and multiple cancers. It has also been found to be very important in AKI and CKD. Although, the authors have tried to put together some relevant information, the article is overall poorly written and contains multiple (too many to list) grammatical and spelling errors. Moreover, the discussion section is completely absent possibly due to wrong version upload. This article needs an extensive revision.

Author Response

We apologize for the grammatical and spelling errors. We have made extensive edits in the revision, and expanded the focus on SFN. All changes are in red font. We hope that this revision will now be acceptable.